# A map of abstract relational knowledge in the human hippocampal–entorhinal cortex

**Mona M Garvert[1,2]\*, Raymond J Dolan[1,3], Timothy EJ Behrens[1,2]**

[1]Wellcome Trust Centre for Neuroimaging, Institute of Neurology, University College London, London, United Kingdom; [2]Oxford Centre for Functional MRI of the Brain, Nuffield Department of Clinical Neurosciences, University of Oxford, Oxford, United Kingdom; [3]Max Planck-UCL Centre for Computational Psychiatry and Ageing Research, London, United Kingdom

**Abstract** The hippocampal–entorhinal system encodes a map of space that guides spatial navigation. Goal-directed behaviour outside of spatial navigation similarly requires a representation of abstract forms of relational knowledge. This information relies on the same neural system, but it is not known whether the organisational principles governing continuous maps may extend to the implicit encoding of discrete, non-spatial graphs. Here, we show that the human hippocampal–entorhinal system can represent relationships between objects using a metric that depends on associative strength. We reconstruct a map-like knowledge structure directly from a hippocampal–entorhinal functional magnetic resonance imaging adaptation signal in a situation where relationships are non-spatial rather than spatial, discrete rather than continuous, and unavailable to conscious awareness. Notably, the measure that best predicted a behavioural signature of implicit knowledge and blood oxygen level-dependent adaptation was a weighted sum of future states, akin to the successor representation that has been proposed to account for place and grid-cell firing patterns.

\*For correspondence: mona. garvert.11@ucl.ac.uk

## Introduction

Animals efficiently extract abstract relationships between landmarks, events, and other types of conceptual information, often from limited experience. Knowing such regularities can help us act in an environment, because the relationships between items that have never been experienced together can easily be computed and exploited in order to make novel inference. In physical space, spatially tuned cells in the hippocampal–entorhinal system have precise place (*O'Keefe and Dostrovsky, 1971*) and grid (*Hafting et al., 2005*) codes that may form the neural basis of a 'cognitive map' (*O'Keefe and Nadel, 1978*). It is likely that the particular form of these representations enables rapid computations of spatial relationships such as distances and vector paths (*Bush et al., 2015*; *Stemmler et al., 2015*). The potential for such rapid online computations embedded into neuronal representations may explain how animals can find novel paths through space (*McNaughton et al., 2006*; *Mittelstaedt and Mittelstaedt, 1980*) or rapidly reroute when obstacles are introduced (*Alvernhe et al., 2011*) or removed (*Alvernhe et al., 2008*). Indeed, in humans, signals that encode distance metrics between landmarks (*Howard et al., 2014*; *Morgan et al., 2011*) and directions to goals (*Chadwick et al., 2015*) can be read out directly from functional magnetic resonance imaging (fMRI) data in the entorhinal cortex.

The hippocampal formation also encodes non-spatial relationships between objects. When these objects can be laid out in a continuous dimension such as time, hippocampal codes extracted from neuronal ensembles (*Rubin et al., 2015*) or fMRI voxel patterns (*Ezzyat and Davachi, 2014*) reflect proximity along this dimension. fMRI signals also appear to reflect veridical angles when two-

**eLife digest** To help us navigate, the brain encodes information about the positions of landmarks in space in a series of maps. These maps are housed by two neighbouring brain regions called the hippocampus and entorhinal cortex. These regions also encode information about non-spatial relationships, for example, between two events that often occur close together in time. However, it was not known whether such non-spatial relationships may also be encoded as a map.

To address this question, Garvert et al. showed volunteers a series of objects on a screen. Unbeknown to the volunteers, the order of the objects was not entirely random. Instead, each object could only follow certain others. The objects were thus connected to one another by a network of non-spatial relationships, broadly comparable to the spatial relationships that connect physical locations in the environment. The next day, the volunteers viewed some of the objects again, this time while lying inside a brain scanner. Although the volunteers still believed that the objects had been presented at random, the activity of their hippocampus and entorhinal cortex reflected the non-spatial relationships volunteers had experienced between the objects. The relationships were organised in an abstract map.

This suggests that the brain organises knowledge about abstract non-spatial relationships into maps comparable to those used to represent spatial relationships. The brain can use these maps of non-spatial relationships to guide our behaviour, even though we have no conscious awareness of the information they contain. The maps may also enable us to make new inferences, just as we can use our spatial maps to find short cuts or navigate around obstacles. Future studies should investigate the mechanisms underlying our ability to create maps of non-spatial relationships and how we use them to guide decision making.

dimensional abstract spaces are formed from continuous dimensions (*Constantinescu et al., 2016*; *Tavares et al., 2015*). However, many relationships that are encoded by the hippocampal formation reflect associations or relationships between discrete objects (*Horner et al., 2015*; *Schapiro et al., 2013*, *2012*; *Wimmer and Shohamy, 2012*). To be a useful source of knowledge, many associations must be organised within an associative structure, but it is unclear how such structures might be represented in the absence of a continuous organising dimension such as space or time.

Highly complex relational structures are often learnt implicitly i.e. unintentionally and without explicit awareness (*Cleeremans et al., 1998*; *Reber, 1989*; *Seger and Augart, 1994*). Neurally, implicitly acquired relational knowledge can be reflected as increases in neural similarity for pairwise associations in the temporal cortex (*Schapiro et al., 2012*) or for members of a temporal community structure (*Schapiro et al., 2013*). However, it is unclear whether map- or graph-like knowledge structures might be encoded non-consciously, i.e. without subjects being aware of relationships between objects.

Here, we explicitly tested this notion using a fMRI adaptation paradigm that allowed us to quantify the relationships between object representations in a neuronal representational space following an implicit learning paradigm. We presented human participants with sequences of objects where stimulus transitions were drawn from random walks along a graph structure. Within the hippocampal-entorhinal system, a map-like organisation of the relationships between object representations could be extracted from fMRI adaptation data acquired on the subsequent day. In this map, associative distance between objects formed a metric that allowed us to extract organising dimensions. This suggests that the brain can automatically organise abstract relational information into map-like structures even if the relationships between objects are non-spatial rather than spatial, discrete rather than continuous, and unavailable to conscious awareness. A signature of map-like encoding was also present behaviourally in a separate group of subjects, demonstrating implicit memory of the structure.

We found no evidence for a mapping of discrete relationships into Euclidian space. Instead, the fMRI adaptation pattern as well as behaviour are more consistent with distance measures reflecting the distribution of future states. These principles are consistent with a predictive representation such as the successor representation (*Dayan, 1993*). Such a predictive map of state space may facilitate

the rapid computation of values in a reinforcement learning world (*Momennejad et al., 2016*; *Russek et al., 2016*). It has recently been demonstrated that the successor representation can account for a number of properties of place cell and grid cell activity (*Stachenfeld et al., 2016*, *2014*).

## Results

We exposed 23 human participants to object sequences whose stimulus transitions, unbeknownst to them, were determined by a random walk in a graph (*Figure 1A*). Subjects performed a behavioural cover task, in which they learned to associate a random stimulus orientation with a button press. In the task instructions, any reference to a sequence or an underlying structure was avoided. After the fMRI experiment, subjects were debriefed and none reported any explicit knowledge of structure in the task. To test whether this exposure to object sequences induced implicit knowledge about the graph, we scanned the subjects on a subsequent day using fMRI while exposing them to a subset of the same objects presented in a random order (only a reduced graph was presented to increase statistical power; *Figure 1B*). In 10% of the fMRI trials, subjects performed an unrelated cover task, reporting whether a grey patch had been present on the preceding object. Neither accuracy nor

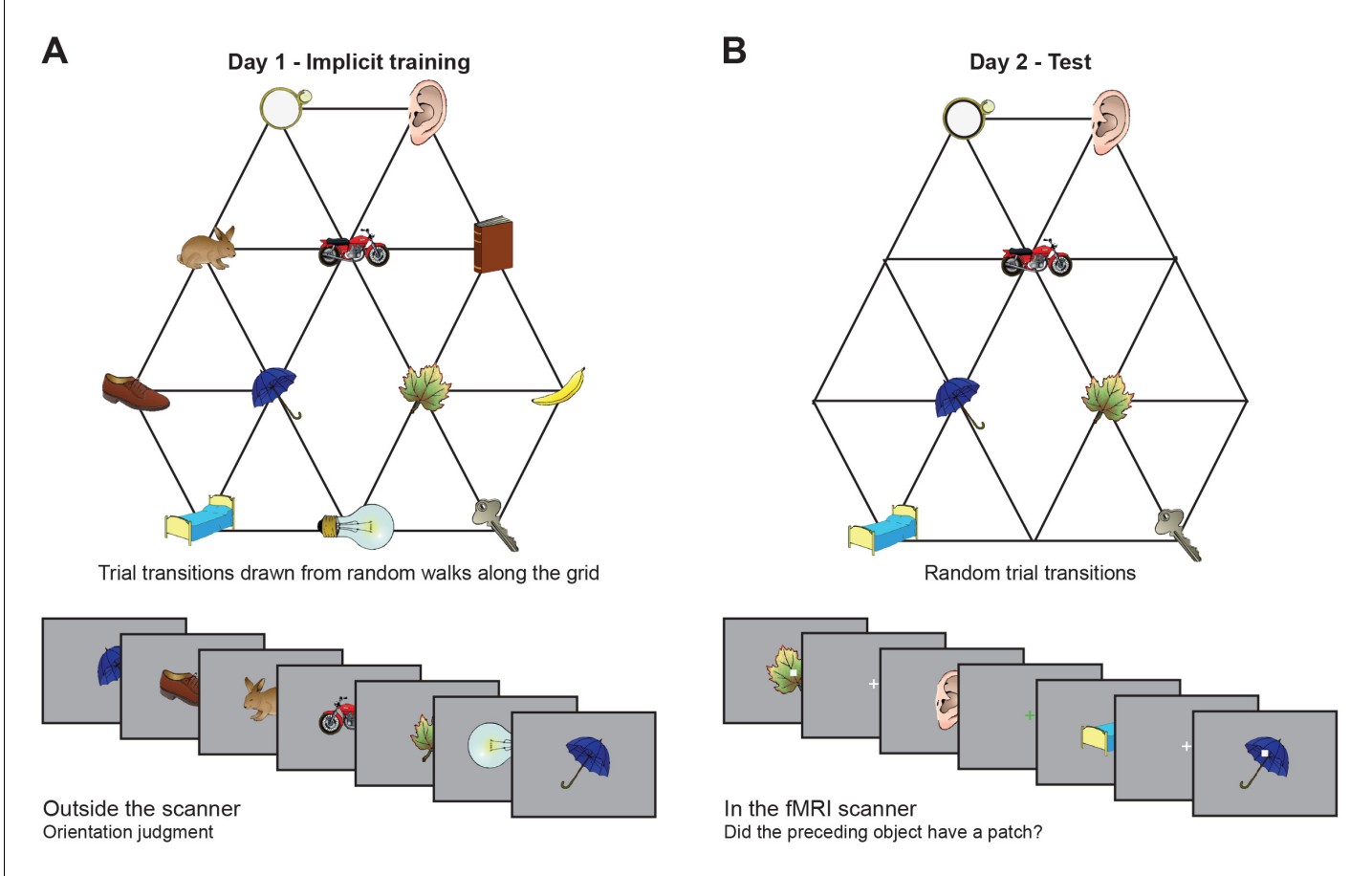

**Figure 1.** Experimental design. (A) Graph structure used to generate stimulus sequences on day 1. Trial transitions were drawn from random walks along the graph. (B) Objects on reduced graph presented to subjects in the scanner on day 2. Trial transitions were random. In both sessions, participants performed simple behavioural cover tasks. See *Figure 1—figure supplement 1* for behavioural performance during the training and the scan sessions. fMRI: functional magnetic resonance imaging.

The following figure supplement is available for figure 1:

**Figure supplement 1.** Task performance.

response time in this task depended on the object on screen or the transition structure (*Figure 1—figure supplement 1*).

We exploited fMRI adaptation (*Barron et al., 2016*; *Grill-Spector et al., 2006*) to investigate the representational similarity for different objects on the graph. We reasoned that in regions encoding a map-like representation of the overall task structure, the degree of similarity in neural representation, and therefore the fMRI adaptation, should decrease as a function of distance between items on the graph. Based on this reasoning, we first looked for brain regions whose fMRI responses to each object increased as a linear function of the link distance of the preceding item.

This adaptation analysis revealed a cluster bilaterally in the entorhinal cortex (*Figure 2A*, family-wise error corrected at peak level within a bilateral entorhinal cortex/subiculum mask, left p=0.014, peak $t_{22}$ = 4.42, [−18, −19, −22] and right p=0.006, peak $t_{22}$ = 4.75, [24, −25, −22]. A right, but not left peak also survived small volume correction (SVC) for a larger region of interest [ROI] comprising the hippocampus, parahippocampal cortex, and entorhinal cortex, left p=0.058 and right p=0.026, see ROIs in *Figure 2—figure supplement 1*). This adaptation effect cannot be explained by basic statistics of the object sequence, such as the number of times an object occurred during training, or basic features of the graph structure, such as the number of neighbours an object has on the graph (*Figure 2—figure supplement 3*).

To confirm the statistical robustness of the effect, and to test whether the effect reflected a gradual increase with distance, we separated the effect into two orthogonal components. These components comprised the difference between connected links (length 1) and all other transitions (lengths 2 and 3; *Figure 2B*, green), and the difference between transitions of length 2 and those of length 3 (*Figure 2B*, red). These two independent contrasts were used to define ROIs bilaterally in overlapping regions of the entorhinal cortex (both thresholded at p<0.01 uncorrected; ROI 1: left peak $t_{22}$ = 3.85; [−18, −19, −22] and right peak $t_{22}$ = 3.26; [24, −25, −22], ROI 2: left peak $t_{22}$ = 4.55, [18, −16, −25] and right peak $t_{22}$ = 3.38, [−18, −25, −25]). Because of their statistical independence, we could use the ROI from one contrast to extract data for the corollary test ($t_{22}$ = 2.27, p=0.03 for length 2 vs. length 3 in ROI 1, *Figure 2C*; and $t_{22}$ = 2.34, p=0.03 for connected vs. all other links in ROI 2, *Figure 2D*). This pair of tests suggests that the fMRI adaptation faithfully represents the link distance. These tests obviate questions of multiple comparisons, because in each case the data are selected from one contrast, and an orthogonal contrast was used for the test statistic.

To further demonstrate this within a single test, we required a coordinate that was independent of all the data. We chose a peak location from an independent study investigating a similar relational measure in spatial maps (*Chadwick et al., 2015*). Extracting data from this coordinate (ROI 3) revealed a linear effect of link distance (*Figure 2E*, $F_{2,44}$ = 10.04, p<0.001), and correspondingly a significant difference between distances of lengths 1 and 3 ($t_{22}$ = 3.71, p=0.001) and lengths 2 and 3 ($t_{22}$ = 3.19, p=0.004), but not between distances of lengths 1 and 2 ($t_{22}$ = 1.67, p=0.11).

Although this distance effect is suggestive of a map-like organisation, it might also merely reflect the temporal proximity between two objects during training. When the temporal and distance relationships between pairs of objects were allowed to compete for variance in a multiple linear regression, the number of links ($t_{22}$ = 3.29, p=0.003), but not time ($t_{22}$ = 1.27, p=0.22), explained the neural signal extracted from the independently defined ROI 3 (*Figure 3A*). Furthermore, relationships between items arranged in a map-like structure are non-directional. Our subjects were not constrained to experience each pair of transitions an equal number of times (*Figure 3B*). Based upon this, we could test whether the fMRI signal was better predicted by the true or symmetrised distance between any two objects. We constructed a measure of the shortest path between each pair of objects according to the actual number of times each transition was experienced by a subject during training (see 'Materials and methods' section). When allowing this measure to compete with its symmetrised, and thereby non-directional, self in a linear model, it was the symmetrised version alone that predicted the fMRI suppression effect (*Figure 3C*, $t_{22}$ = 2.78, p=0.01 and $t_{22}$ = −1.64, p=0.11).

In order to test whether these map-like features are a consequence of a map-like organisation, we organised the signal into a 7 × 7 matrix, with each matrix element reflecting the mean fMRI response across subjects to transitions between the corresponding pairs of objects (*Figure 3D*). For example, element [2,7] in this matrix is the response to object 7 when preceded by object 2 on the graph, averaged across all subjects. Because the signal is suppressed for nearby objects, this matrix is analogous to a distance matrix. When we applied multidimensional scaling (MDS) in order to visualise the most faithful two-dimensional representation of distances in this matrix, the graph structure

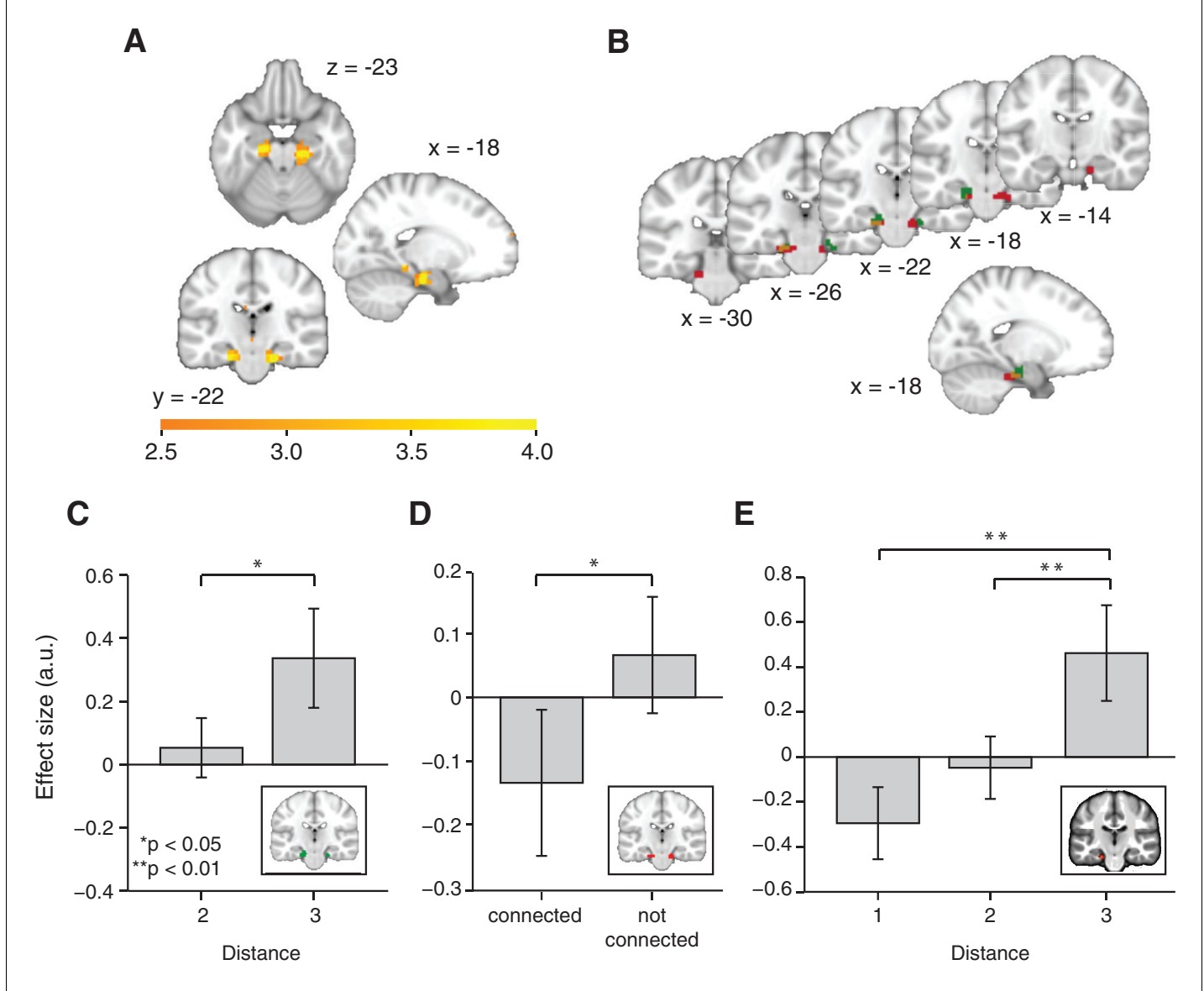

**Figure 2.** Functional magnetic resonance imaging adaptation in the hippocampal–entorhinal system decreases with distance on the graph. (A) Whole-brain analysis showing a decrease in functional magnetic resonance imaging adaptation with link distance in the hippocampal–entorhinal system, thresholded at p<0.01, uncorrected for visualisation. (B) Within the hippocampal–entorhinal system, green indicates greater suppression if the preceding stimulus was a neighbour relative to a stimulus two or three links away. Red indicates greater suppression if a preceding stimulus was two links away than three links away. The depicted areas were used as regions of interest for analyses in (C) (green) and (D) (red). (C) Parameter estimates for link 2 versus link 3 transitions extracted from the green entorhinal region of interest in *Figure 2B* (t$_{22}$ = 2.27, p=0.03). Other brain areas do not show this increase in activity with distance (*Figure 2—figure supplements 2D,E*). (D) Parameter estimates extracted from the red entorhinal region of interest in *Figure 2B*, sorted according to whether objects were connected on the graph or not (t$_{22}$ = 2.34, p=0.03). (E) Parameter estimates extracted from the peak MNI coordinate reported in *Chadwick et al. (2015)*, [−20, −25, −24] and sorted according to distance (F$_{2,44}$ = 10.04, p=0.0003). See *Figure 2—figure supplement 1* for masks used for small-volume correction in *Figure 2A, Figure 2—figure supplement 2* for distance-dependent scaling effects in other brain regions and *Figure 2—figure supplement 3* for effects of object familiarity and centrality. Error bars show mean and standard error of the mean. a.u.: arbitrary units.

The following figure supplements are available for figure 2:

**Figure supplement 1.** Anatomically defined regions of interest used for small-volume correction.

**Figure supplement 2.** Distance-dependent scaling of neural activity is specific to the hippocampal–entorhinal system.

*Figure 2 continued on next page*

*Figure 2 continued*

**Figure supplement 3.** Effects of object familiarity.

of our experimental map was recovered despite the subjects' professed ignorance of any such organisation (*Figure 3E*). Permutation tests confirm that the multidimensional scaling-mapped distances are significantly more correlated with link distances of the original graph structure than with link distances of a null distribution consisting of all other complete graphs with seven links (r = 0.65, p=0.003, *Figure 3—figure supplement 2A*). Furthermore, no links cross in the graph resulting from the MDS mapping. This is only true for 13.17% of all possible graphs with nodes in the same location, but seven randomly distributed links (*Figure 3—figure supplement 2B*). Notably, the data were extracted from an independent ROI taken from an experiment investigating maps in allocentric physical space (*Chadwick et al., 2015*, ROI 3). Results are comparable if parameter estimates are extracted from an anatomically defined ROI comprising the entorhinal cortex and the subiculum (*Figure 3—figure supplement 3*).

In the reinforcement learning literature, it has been suggested that a cognitive map of the relationship between states may be most useful if the representation of a state is predictive in nature and reflects the distribution of likely future states. This idea has been formalised as the successor representation (*Dayan, 1993*; *Momennejad et al., 2016*; *Russek et al., 2016*), proposed to be encoded by hippocampal place cells (*Stachenfeld et al., 2016*, *2014*). According to this view, hippocampal place cells do not encode an animal's current location in space, but instead encode a predictive representation of future locations. The successor representation may facilitate reinforcement learning, because the resulting predictive measure of future states could be flexibly combined with reward representations to enable rapid computation of navigational trajectories (*Baram et al., 2017*; *Dayan, 1993*; *Momennejad et al., 2016*; *Russek et al., 2016*).

Mathematically, the successor representation can be computed from the adjacency matrix $A$ that defines the relationship between states:

$$\sum_{n=0}^{\infty} \gamma^n A^n = (I - \gamma A)^{-1}$$

with a discount factor $\gamma < 1$. Here, entries $a_{ij}$ for each $A^n$ correspond to the number of possible paths of length $n$ between objects $i$ and $j$. The successor representation therefore computes the weighted sum of distant future states, with $A^n$ discounted more heavily for larger $n$ (i.e. for longer paths between pairs of objects).

Notably, this same representation is common in graph theory, where the matrix $(I - \gamma A)^{-1}$ is termed the matrix resolvent and is used to measure the proximity or 'communicability' between nodes in the graph. Graph theory also proposes a second measure that is closely related, the matrix exponential (*Estrada and Hatano, 2008*, *2010*):

$$e^A = \sum_{n=0}^{\infty} \frac{A^n}{n!}$$

Both measures compute a weighted sum over future states, which easily generalises from continuous to discrete, and from two-dimensional to high-dimensional spaces.

In order to test whether the neural distance effects are consistent with such predictive measures, we tested for areas whose fMRI responses to each object increased as a linear function of communicability, corresponding to the negative of the matrix exponential. Compared to the successor representation, the matrix exponential has the advantage that it does not require the fitting of a free parameter. The matrix exponential is small for nodes that are far away from each other on a graph, such that it scales negatively with distance. Unlike a mapping into Euclidian space, communicability significantly distorts the graph structure by shortening links that form part of many paths around the graph structure and lengthening links that would be less frequently visited by a random navigator (*Figure 4A*).

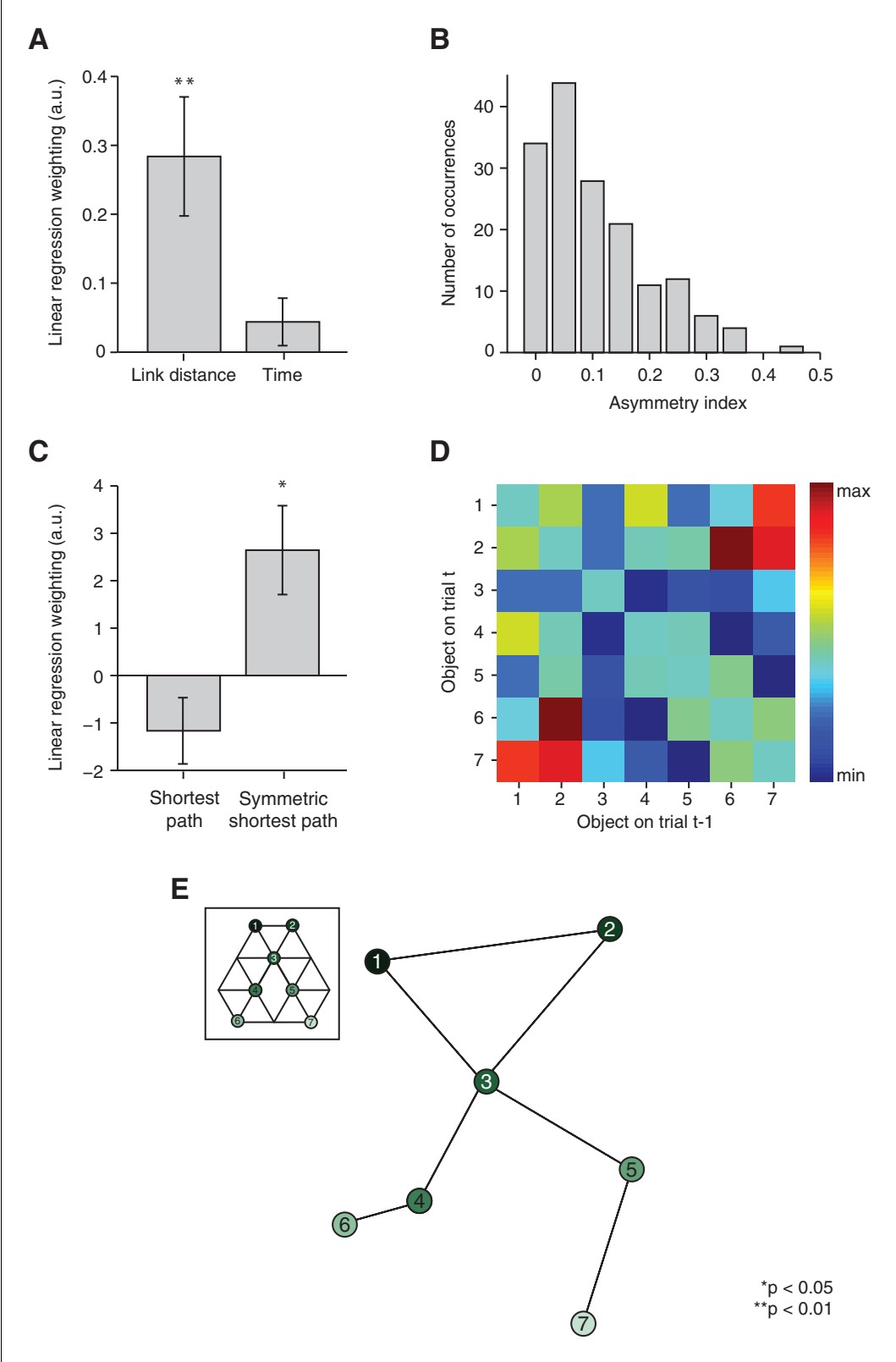

**Figure 3.** Relational information is organised as a map. (A) Linear regression on neural activity with number of links and average time between two objects during training as regressors ($t_{22}$ = 3.29, p=0.003 and $t_{22}$ = 1.27, p=0.22). (B) Absolute difference in the number of times a transition was visited in one versus the other direction (e.g. 5 preceded by 1 vs. 1 preceded by 5) normalised by the total number of visits in either direction for all subjects. (C) Multiple linear regression on neural activity with the shortest path between objects and the symmetrised shortest path between objects as

*Figure 3 continued on next page*

*Figure 3 continued*

regressors (t$_{22}$ = −1.64, p=0.11 and t$_{22}$ = 2.78, p=0.01). (**D**) 7 × 7 matrix representing the average fMRI signal in response to an object depending on which other object preceded, averaged across subjects and symmetrised. Objects were never repeated during scanning; the diagonal entries are therefore set to 0. This matrix was used for the multidimensional scaling visualised in (**E**). (**E**) Visualisation of the localisation of the object representations in a two-dimensional space according to multidimensional scaling. Lines indicate transitions experienced during training. The distances between the resulting locations of nodes in a two-dimensional space are significantly correlated with the link distances of the original graph structure (r = 0.65, p=0.003, see *Figure 3—figure supplement 2* for the null distribution used for the permutation test). All analyses were performed on data extracted from the peak MNI coordinate reported in *Chadwick et al. [2015]*), [−20, −25, −24], but also hold in an anatomically defined region of interest including the entorhinal cortex and the subiculum (*Figure 3—figure supplement 3*). See *Figure 3—figure supplement 1* for results if object-specific activity is removed. Error bars show mean and standard error of the mean. a.u.: arbitrary units.

The following figure supplements are available for figure 3:

**Figure supplement 1.** The distance-dependent scaling cannot be driven by a main effect of object position.

**Figure supplement 2.** Map characteristics in a null distribution generated by permuting the links making up the graph structure.

**Figure supplement 3.** Distance effects in an anatomically defined region of interest comprising the entorhinal cortex and the subiculum.

We find that neural activity bilaterally in the hippocampal–entorhinal system scales with communicability (*Figure 4B, C*, family-wise error corrected at peak level within a bilateral entorhinal cortex/subiculum mask, left p=0.001, peak t$_{22}$ = 5.47 [−18, −19, −25] and right p=0.0005, peak t$_{22}$ = 5.79, [21, −19, −28]). Both clusters also survived SVC (small-volume correction) for a larger ROI comprising the hippocampus, parahippocampal cortex, and entorhinal cortex (left p=0.004 and right p=0.004, see ROIs in *Figure 2—figure supplement 1B*). Activity in the same areas also scales with the negative of the successor representation if the free parameter $\gamma$ is set to the commonly used value of $\gamma = \frac{0.85}{\lambda_{max}}$ ($\lambda_{max}$ = largest eigenvalue of $A$ in modulus, *Aprahamian et al., 2016*; *Benzi and Klymko, 2013*, *Figure 4—figure supplement 1*).

In the left hippocampal formation, communicability effects are significant even if Euclidian distances are included as an additional regressor (*Figure 4D*, p=0.006, peak t$_{22}$ = 4.72, [−15, −13, −19], SVC mask 1 and p=0.027, SVC mask 2). This suggests that the hippocampal–entorhinal system does not map the graph structure into a Euclidian space. Instead, these results are consistent with the view that the distance effect we observe in this system may be a consequence of the hippocampal formation encoding a predictive representation of states within a graph structure (*Stachenfeld et al., 2016, 2014*).

A map-like representation in the hippocampal–entorhinal system suggests that subjects acquired implicit knowledge about the graph structure, even in the absence of explicit awareness of any regularities in the object sequence. To reveal such implicit learning behaviourally, we asked an independent group of 26 participants, who were trained in the same way as the scanning cohort on the first graph structure (*Figure 5A*), to repeat the object orientation cover task on day 2. As was the case for scanned subjects, object transitions were now random and only objects from a reduced graph were presented (*Figure 5B*). We hypothesised that implicit knowledge about the graph structure would influence response times, such that subjects would respond faster if a preceding object in the test sequence was closer on the graph structure underlying the train sequence. Indeed, we found that log-transformed response times were longer the further away the preceding object was on the graph (*Figure 5C, D*). In line with our imaging results, response times did not scale with link or Euclidian distance between objects, but instead with communicability (communicability: t$_{25}$ = 2.77, p=0.01; link distance: t$_{25}$ = −0.40, p=0.69; Euclidian distance: t$_{25}$ = −0.85, p=0.40, *Figure 5C, D*).

## Discussion

The hippocampal–entorhinal system is engaged when an animal navigates within a physical environment and acquires flexible knowledge about spatial relationships. In mammals, the hippocampal–entorhinal system contributes to spatial navigation by mapping relationships in situations where knowledge is physical, continuous, and consciously available (*Chadwick et al., 2015*;

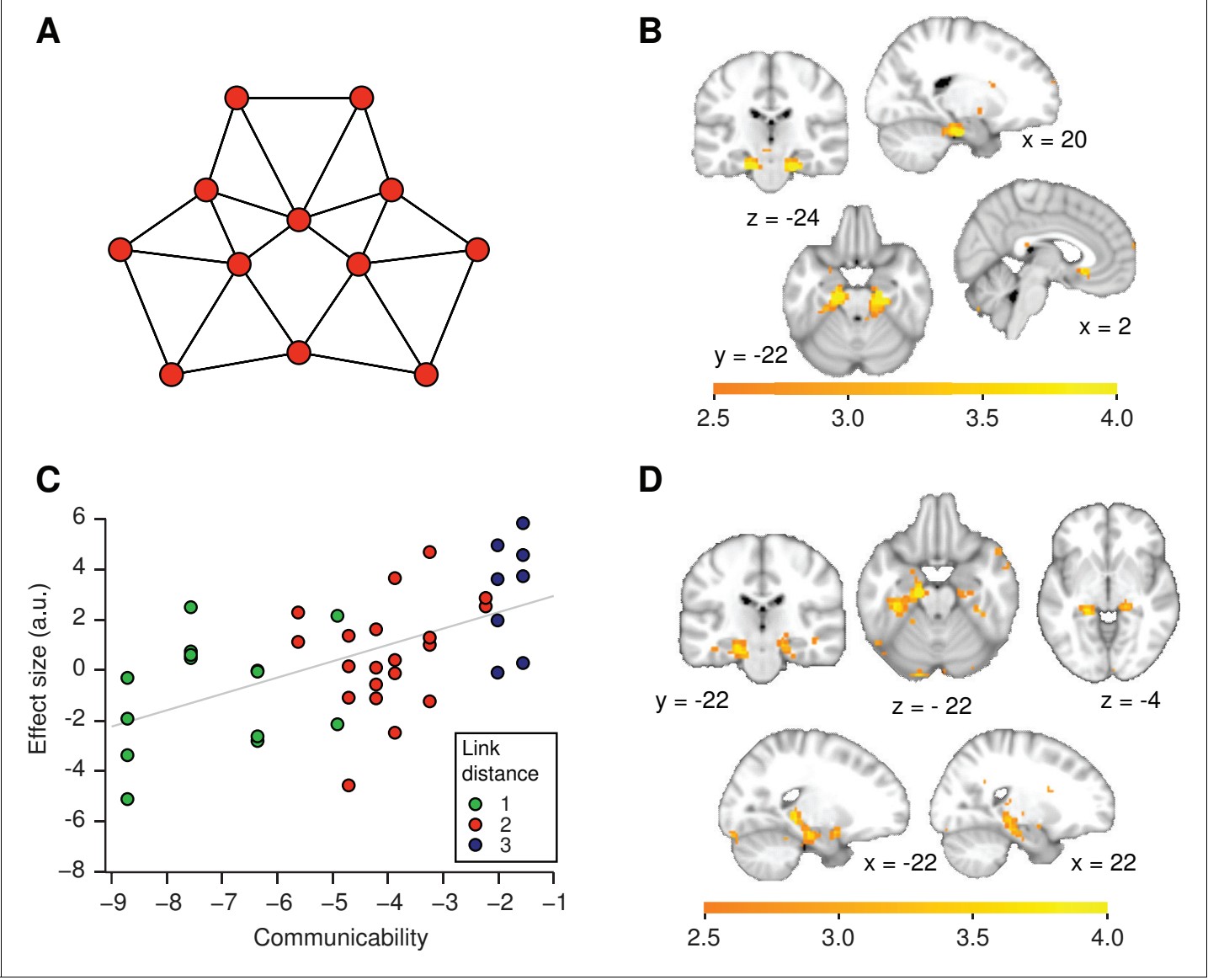

**Figure 4.** Functional magnetic resonance imaging adaptation in the hippocampal–entorhinal system is consistent with predictive representations of relational knowledge. (A) Visualisation of communicability coordinates for the graph structure by performing multidimensional scaling on the communicability matrix. (B) Whole-brain regression of communicability onto neural activity. (C) Visualisation of the communicability effect. Average parameter estimate for each of the 42 stimulus transitions across subjects extracted from a bilateral region of interest in (B) (thresholded at p<0.01). The colours of the dots correspond to link distances. This graph is added for visualisation purposes only as the parameter selection is biased. (D) Whole-brain regression of communicability onto neural activity when Euclidian distances are included as an additional regressor in the general linear model. All statistical maps are thresholded at p<0.01 for visualisation. a.u.: arbitrary units.

The following figure supplement is available for figure 4:

**Figure supplement 1.** Activity in the hippocampal–entorhinal system is consistent with the successor representation.

*Derdikman and Moser, 2010*; *Howard et al., 2014*; *Spiers and Maguire, 2007*). Here, we used a statistical learning paradigm to demonstrate that the hippocampal–entorhinal system also efficiently extracts statistical regularities in a non-spatial task where the relationships between items are discrete, and organises this non-spatial relational knowledge in an abstract relational map, suggesting that the hippocampal–entorhinal system creates metric representations of discrete relationships

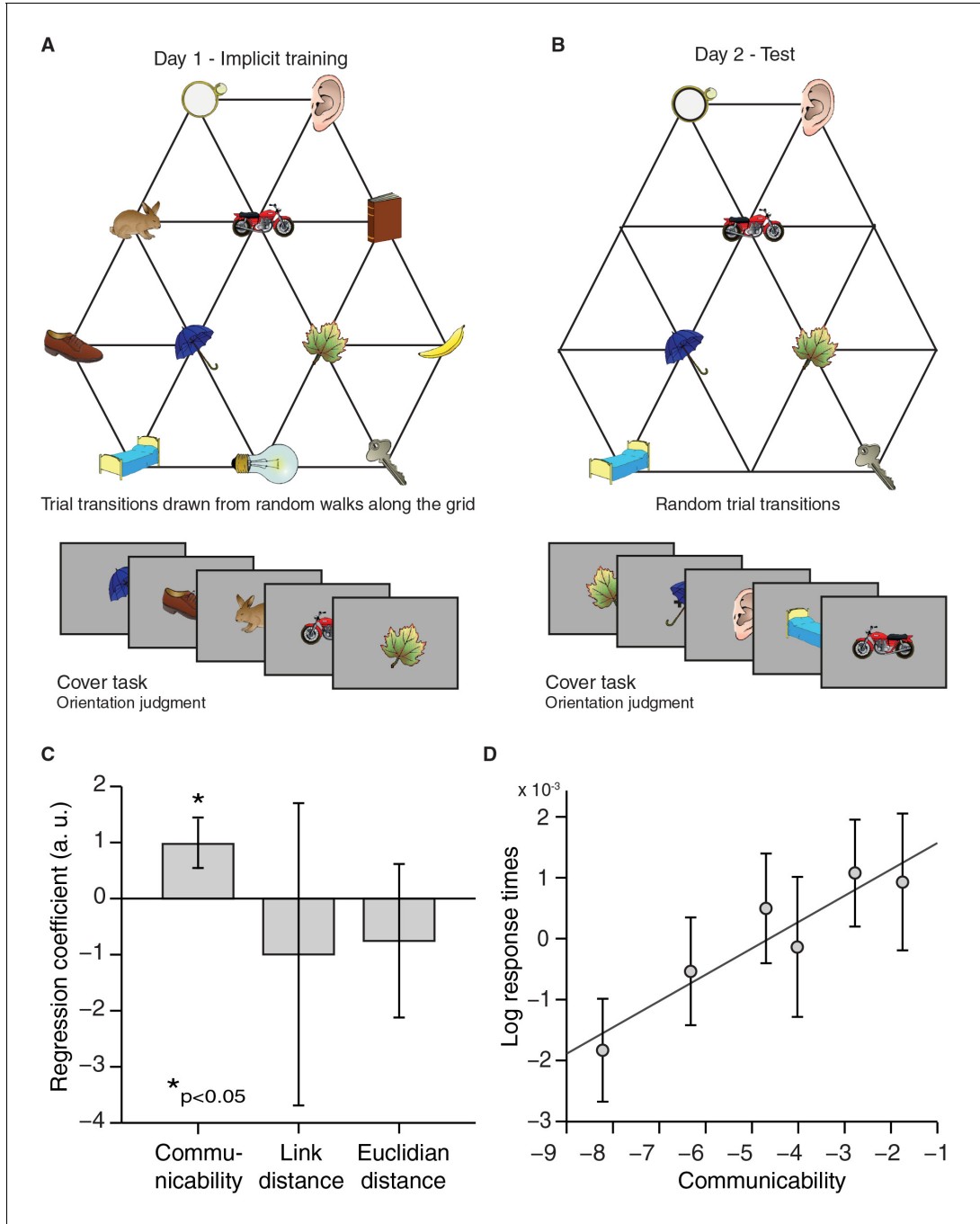

**Figure 5.** Response times reflect graph structure. (A) Graph structure used to generate stimulus sequences on day 1. Trial transitions were drawn from random walks along the graph structure. (B) Objects on reduced graph presented to subjects on day 2. Trial transitions were random. In both sessions, participants performed an object orientation cover task under which response times were measured. (C) A regression of communicability, link distance, and Euclidian distance onto log-transformed response times across subjects (communicability: $t_{25}$ = 2.77, p=0.01; link distance: $t_{25}$ = −0.40, p=0.69; Euclidian distance: $t_{25}$ = −0.85, p=0.40). (D) Visualisation of the relationship between communicability and log-transformed response times. Communicability measures were divided into six bins with an equal number of object–object transitions per bin. The y-axis corresponds to the average demeaned log-response time across subjects for each bin. Error bars denote the standard error of the mean. a.u.: arbitrary units.

based on associative strengths (*Eichenbaum and Cohen, 2014*). Notably, there were no continuous dimensions in our discrete stimulus sets. The dimensions had to be extracted from the associations.

These results add to the notion that the hippocampal formation maps experiences across a wide range of different dimensions, thereby supporting flexible behaviour across many domains of life (*O'Keefe and Nadel, 1978*; *Tolman, 1948*). Recent studies have focused on the human hippocampal formation for storing the relational knowledge that makes up simple world models in reinforcement learning (*Barron et al., 2013*; *Boorman et al., 2016*; *Bornstein and Daw, 2013*; *Wimmer and Shohamy, 2012*). Our findings extend these simple associative paradigms to more complex associative maps, and demonstrate that these maps may be learnt implicitly.

Notably, the resulting map is not Euclidian in nature. Instead, we find that the neural data as well as the behaviour are consistent with a measure corresponding to a weighted sum of future states in the graph structure. This is consistent with the idea that reinforcement learning benefits from knowledge about warped geometries in space, which may be represented as predictive maps, or successor representations, in the hippocampal formation (*Stachenfeld et al., 2016*, *2014*). Such predictive representations of the relationships between discrete objects or states of the world may be combined with reward representations to enable flexible goal-directed behaviour (*Baram et al., 2017*; *Dayan, 1993*; *Momennejad et al., 2016*; *Russek et al., 2016*).

Our data suggest that the map of the relational structure in discrete abstract space may be encoded by the pattern of firing in the entorhinal cortex, which also encodes maps in continuous physical space. The entorhinal cortex is noted for the presence of grid cells (*Hafting et al., 2005*), which may provide a metric for measuring distances in physical space (*Bush et al., 2015*; *Stemmler et al., 2015*), allowing vector navigation. However, recent theoretical treatments suggest that grid-like firing might also be understood as a principal component decomposition of the covariance information between place cells (*Dordek et al., 2016*) or the transitions between states (*Stachenfeld et al., 2016*, *2014*). Together with the finding of hippocampal cells in humans that encode individual concepts (which may be analogous to place cells), these theories can explain how grid-like firing patterns could extend to discrete spaces (such as the one we have used here), and also make predictions for how grid-like coding might extend to higher-dimensional spaces (*Baram et al., 2017*).

Our behavioural results are consistent with the observation that the human brain can acquire abstract knowledge unconsciously and automatically by extracting statistical regularities from incidental exposure to events generated according to a set of rules (*Berry and Broadbent, 1984*; *Cleeremans et al., 1998*; *Reber, 1989*, *1967*; *Seger and Augart, 1994*). The resulting implicit knowledge can be used to guide behaviour despite an inability to verbalise the underlying regularities. Neurally, implicit learning of sequences involves medial temporal lobe structures, including the hippocampus, subiculum, and entorhinal cortex (*Schendan et al., 2003*). The hippocampal–entorhinal system also responds to the violation of learnt sequence structures (*Kumaran and Maguire, 2006*) and signals the likelihood of events in learnt sequences (*Strange et al., 2005*). This may be facilitated by an increase in representational similarity as stimuli become embedded into knowledge structures (*Schapiro et al., 2013*, *2012*). Our results add to this literature by proposing a specific way in which implicit knowledge may be organised in maps to facilitate goal-directed behaviour.

It is notable that we did not find clear evidence for the map-like structure outside the hippocampal formation, although areas such as the orbitofrontal cortex have been shown to represent cognitive maps of decision spaces (*Schuck et al., 2016*). It is possible that this is because our study relies on implicit learning, and because the subjects do not have to use the associative structure for any task. Indeed, we note that neural signals can be recorded in frontal and parietal cortices, reflecting the 'state-prediction errors' that ensue when predicted state relationships are breached during behavioural control (*Gläscher et al., 2010*). Similar prediction errors in the orbitofrontal cortex during active learning predict later changes in hippocampal representations of the stored model (*Boorman et al., 2016*). These and similar ideas have led to a theoretical account of place and grid activity in the hippocampal formation as state representations in reinforcement learning models (*Stachenfeld et al., 2016*, *2014*).

It has long been known that the hippocampal formation is important for tasks that rely on associative and relational knowledge. It supports the organisation of stimuli across arbitrary stimulus dimensions such as temporal co-occurrence (*Schapiro et al., 2013*, *2012*) or social rank (*Kumaran et al., 2012*), and organises behaviourally relevant stimulus categories in a hierarchy (*McKenzie et al.,*

*2014*). These organisational principles facilitate generalising over individual episodes (*Komorowski et al., 2013*) and enable transitive inference by combining newly formed associations between discrete stimuli (*Collin et al., 2015*; *Heckers et al., 2004*; *Horner et al., 2015*; *Preston et al., 2004*; *Schlichting et al., 2015*). Value spreading across associated stimulus representations in the hippocampus can then directly influence behaviour in novel decision-making situations (*Wimmer and Shohamy, 2012*). We hope that the current findings help to reconcile these results with the spatial functions of the same neural structures. Such an organisation of relational information might be the basis for an animal's ability to navigate through an abstract concept space and to perform flexible computations without direct experience.

## Materials and methods

### Subjects

Twenty three volunteers (aged 18–31 years, mean age ± standard deviation 23.5 ± 3.7 years, 15 males) with normal or corrected-to-normal vision and no history of neurological or psychiatric disorders participated in the fMRI experiment. All subjects gave written informed consent and the study was approved by the University College London Hospitals Ethics Committee. The study took place at the Wellcome Trust Centre for Neuroimaging. Subjects were naïve to the purpose of the experiment.

### Stimuli and task

Thirty one coloured and shaded object images that were similar in terms of their familiarity and complexity were selected from the 'Snodgrass and Vanderwart 'Like' Objects' picture set (http://wiki.cnbc.cmu.edu/Objects, *Rossion and Pourtois, 2004*). For each subject, a subset of 12 objects was chosen and randomly assigned to the 12 nodes of the graph shown in *Figure 1A*. On day 1, subjects were exposed to object sequences generated from a random walk on the graph, where only objects that were directly connected to another object by a link could follow a presentation of this object. To avoid local repetitions, we constrained sequences such that at least three objects had to occur between any two presentations of the same object. Each object was randomly presented in one of two orientations, which were mirror images of each other.

Before the start of the experiment, subjects were shown the entire set of 12 stimuli and instructed to remember which of two buttons to press for a particular object orientation (normal or mirrored). During the actual training, subjects were instructed to press the button associated with the stimulus orientation as quickly and accurately as possible while watching the object sequences. Visual feedback after each button press indicated whether or not a response was correct. Object orientation was randomised across trials and key assignment was counterbalanced across subjects. Subjects learnt to perform the task quickly and accurately (*Figure 1—figure supplement 1*). Stimuli were presented for 2 s and each experimental block consisted of 133 object presentations. Subjects performed this experiment for 12 blocks in total. Between blocks (ca. every 5 min), subjects were free to take self-paced breaks.

On the next day, subjects were presented with object sequences in the scanner. Only the seven objects corresponding to the locations illustrated in *Figure 1B* were presented and stimuli were never repeated. This reduced the total number of stimulus–stimulus transitions and thereby increased statistical power for our key question of interest, as this large number of times that each transition was probed provided us with a more accurate estimate of the respective suppression effects. Furthermore, stimulus transitions did not follow the graph structure, but were instead randomised with a constraint that each of the 42 possible object transitions occurred exactly 10 times per block (objects were never repeated).

The fMRI experiment consisted of 421 items per run and three experimental runs. Stimuli were presented for 1 s, with a jittered inter-trial interval (ITI) generated from a truncated Poisson distribution with a mean of 2 s. While observing the object sequences, subjects performed a cover task of infrequently reporting by button press whether a small grey patch had appeared on a preceding trial. The patch was present on a randomly selected 50% of the objects. Trials on which subjects had to report the existence of a grey patch were signalled by a green cross during the inter-stimulus interval instead of the standard white cross. The cross was green exactly once after each of

the 42 possible transitions (i.e. in 10% of the total number of trials). In 50% of those cases, a patch had been present on the preceding trial. Each correct button press was rewarded with £0.10 paid out in addition to a £33 show-up fee to ensure that subjects attended to the stimuli. Subjects received brief training on this task before they performed it in the scanner. Key assignment was counterbalanced across subjects. Subjects performed the cover task very well (correct performance rate across subjects: 94 ± 3%, mean ± standard error of the mean), confirming that they paid attention to the presented objects throughout the duration of the scan.

After the experiment, subjects were asked whether they noticed any differences between the object sequences presented on day 1 and the object sequences presented in the scanner on day 2. While subjects realised that some objects were missing on day 2, none reported any awareness of the fact that the sequence differed in any other way. When asked explicitly, no subject was aware of the fact that the sequences on day 1 were generated according to an underlying structure.

## fMRI data acquisition and pre-processing

Visual stimuli were projected onto a screen via a computer monitor. Subjects indicated their choice using an MRI-compatible button box.

MRI data were acquired using a 32-channel head coil on a 3 Tesla Allegra scanner (Siemens, Erlangen, Germany). A T2*-weighted echo-planar sequence was used to collect 43 transverse slices (ascending order) of 2-mm thickness with 1-mm gaps and an in-plane resolution of 3 × 3 mm, a repetition time of 3.01 s, and an echo time of 70 ms. Slices were tilted by 30° relative to the rostro-caudal axis and a local z-shim with a moment of −0.4 mT/m was applied to the orbitofrontal cortex region (*Weiskopf et al., 2006*). The first five volumes of each block were discarded to allow for scanner equilibration. After the experimental sessions, a T1-weighted anatomical scan with 1 × 1 × 1 mm resolution was acquired. In addition, a whole-brain field map with dual echo-time images (TE1 = 10 ms, TE2 = 14.76 ms, resolution 3 × 3 × 3 mm) was obtained in order to measure and later correct for geometric distortions due to susceptibility-induced field inhomogeneities.

We performed slice time correction, corrected for signal bias, and realigned functional scans to the first volume in the sequence using a six-parameter rigid body transformation to correct for motion. Images were then spatially normalised by warping subject-specific images to an MNI (Montreal Neurological Institute) reference brain, and smoothed using an 8-mm full-width at half maximum Gaussian kernel. All pre-processing steps were performed with SPM12 (Wellcome Trust Centre for Neuroimaging, http://www.fil.ion.ucl.ac.uk/spm).

## fMRI data analysis

We implemented three types of event-related general linear models (GLMs) in order to analyse the fMRI data. The first set of GLMs contained separate onset regressors for each of the seven objects with a patch and without a patch. Each onset regressor was accompanied by different parametric regressors. These corresponded to the link distance (defined as the minimum number of links between the pair of items; i.e. distance 1, 2, or 3, *Figure 2A* and *Figure 2—figure supplement 2A*), the communicability (see below, *Figure 4B*), and the negative of the successor representation (*Figure 4—figure supplement 1B*) between the object on trial t and the preceding object on trial t − 1 on the graph presented in *Figure 1B*. For the analysis reported in *Figure 4D*, both communicability and Euclidian distances were included as parametric regressors.

Communicability was computed as the negative of the matrix exponential of the adjacency matrix $A$, describing the relationship between nodes on the graph:

$$c = -e^A = -\sum_{n=0}^{\infty} \frac{A^n}{n!}$$

The successor representation was computed as:

$$\sum_{n=0}^{\infty} \gamma^n A^n = (I - \gamma A)^{-1}$$

with $\gamma$ set to the commonly used value of $\frac{0.85}{\lambda_{max}}$ ($\lambda_{max}$ = largest eigenvalue of $A$ in modulus, *Aprahamian et al., 2016*; *Benzi and Klymko, 2013*).

Euclidian distances were computed from the graph in *Figure 1A*, with all distances between objects connected by a link set to 1.

In the second type of GLM (GLM 2), all 42 possible object transitions (object 1 preceded by object 2; object 1 preceded by object 3, . . ., object 7 preceded by object 6) were modelled separately for patch trials and no-patch trials.

A third type of GLM contained one onset regressor for all objects with a patch, and a separate onset regressor for objects without a patch. Each onset regressor was accompanied by a parametric regressor indicating the number of times an object was presented during training (*Figure 2—figure supplement 3*).

All GLMs included a button press regressor as a regressor of no interest. Trials associated with a button press and the two subsequent trials were not included in the main regressors in order to avoid button press-related artefacts. All regressors were convolved with a canonical haemodynamic response function. Because of the sensitivity of the blood oxygen level-dependent signal to motion and physiological noise, all GLMs also included six motion regressors obtained during realignment, as well as 10 regressors for cardiac phase, 6 for respiratory phase and 1 for respiratory volume extracted with an in-house developed Matlab toolbox as nuisance regressors (*Hutton et al., 2011*). Models for the cardiac and respiratory phase and their aliased harmonics were based on RETROI-COR (*Glover et al., 2000*). Sessions were modelled separately within the GLMs.

The contrast images of all subjects from the first level were analysed as a second-level random effects analysis. We report our results in the hippocampal–entorhinal formation, as this was our *a priori* ROI, at a cluster-defining statistical threshold of $p < 0.001$ uncorrected, combined with SVC for multiple comparisons (peak-level family-wise error [FWE] corrected at $p < 0.05$). For the SVC procedure, we used two different anatomical masks. The first mask consisted of the entorhinal cortex and subiculum alone and was received with thanks from *Chadwick et al. (2015)*, (*Figure 2—figure supplement 1A*). The second mask also contained other medial temporal lobe regions implicated in encoding physical space and comprised the hippocampus, entorhinal cortex, and parahippocampal cortex, as defined according to the maximum probability tissue labels provided by Neuromorphometrics, Inc. (*Figure 2—figure supplement 1B*). Activations in other brain regions were only considered significant at a level of $p < 0.001$ uncorrected if they survived whole-brain FWE correction at the cluster level ($p < 0.05$). While no areas survived this stringent correction for multiple comparisons, other regions are reported in *Figure 2—figure supplement 2* at an uncorrected threshold of $p < 0.01$ for completeness. While we used masks to correct for multiple comparisons in our ROI, all statistical parametric maps presented in the manuscript are unmasked.

To independently test for distance-dependent scaling of activity within the entorhinal cortex, we defined two different ROIs based on two orthogonal contrasts from non-patch trials in GLM 2. The first ROI was defined on the basis of decreased activity in transitions where the preceding object was directly connected with the current object (e.g. regressors corresponding to transition 1–2, 6–4, or 5–7, see *Figure 1—figure supplement 1C*) relative to all other transitions (e.g. regressors corresponding to transition 4–2, 7–4, or 1–7; i.e. non-connected–connected). This contrast revealed that clusters in the bilateral entorhinal cortex show more adaptation if a preceding object is connected with a currently presented object, relative to a situation where the preceding object is 2 or 3 links away (green in *Figure 2B* and *Figure 2—figure supplement 2B*). This defined ROI 1 (thresholded at $p < 0.01$), from which we then extracted parameter estimates for each of the 42 no-patch transitions and tested for an orthogonal distance effect, namely whether activity differed for transitions of distance 2 relative to transitions of distance 3 using a two-tailed paired t-test *Figure 2C*.

In a second independent test, we defined a bilateral entorhinal ROI based on the following contrast: [transitions with 3 links between the relevant objects] − [transitions with 2 links between the relevant objects]. This contrast is orthogonal to the first contrast and identified brain regions that responded more strongly on a trial if the preceding object was 3 links rather than 2 links away (red in *Figure 2B* and *Figure 2—figure supplement 2C*). Again, we extracted parameter estimates for each of the 42 non-patch onset regressors from ROI 2 and performed an orthogonal test for distant-dependent scaling by investigating whether activity in this region was also significantly different for directly connected versus non-connected objects using a two-tailed paired t-test (e.g. transition 1–2, 6–4, or 5–7 versus transition 4–2, 7–4, or 1–7 in *Figure 1—figure supplement 1C*), see *Figure 2D*.

Note that the distance-dependent scaling effects cannot be explained by object-specific differences in activity within these ROIs. While the mean activity for different objects differs slightly, but non-significantly (*Figure 3—figure supplement 1A,C,E*), removing these main effects by subtracting the mean activity for each object before performing the above-described analyses does not alter the results (*Figure 3—figure supplement 1B,D,F*).

In a further independent test of the distance-dependent scaling of activity in the hippocampal–entorhinal system, we extracted parameter estimates from a ROI defined based on an independent study investigating the representation of a geocentric goal direction in the entorhinal/subicular region (ROI 3, *Chadwick et al., 2015*). Specifically, we extracted parameter estimates for the 42 non-patch transitions from the peak voxel reported in his study (MNI coordinates: [−20, −25, −24]). This definition of a ROI was non-biased and allowed us to test directly for distance-dependent scaling of activity. We first performed a repeated-measures analysis of variance and *post-hoc* planned two-tailed paired t-tests on the parameter estimates sorted according to distance (*Figure 2E*). To investigate whether information is organised with respect to the distance relationship or with respect to the average time that passed between the occurrence of two objects during training, we performed a multiple linear regression. In this regression analysis, we included one regressor denoting the distance between object pairs on the graph (1, 2, and 3) and a second regressor accounting for the average number of objects that had occurred between any pair of objects $i$ and $j$ during training. Since the duration of object presentations and the ITI during training were constant, this measure was directly proportional to the time elapsed between the occurrence of the two objects. The dependent variable in the regression analysis was the neural activity for the 42 non-patch transition regressors extracted from this independently defined peak voxel (ROI 3, *Figure 3A*). To assess the significance across subjects, we performed two-tailed paired t-tests on the regression coefficients.

To test for the directionality of the distance effect in the entorhinal cortex, we exploited the fact that subjects were not exposed to transitions between connected objects in the two directions (e.g. 5 followed by 3 vs. 3 followed by 5) equally often. To assess the variability in the number of times a transition was experienced in one versus the other direction during training, we defined an asymmetry index as:

$$a = \frac{|xy - yx|}{xy + yx}$$

where $xy$ corresponds to the number of times object $y$ was preceded by object $x$ during training and $yx$ corresponds to the number of times object $x$ was preceded by object $y$ during training. An asymmetry index of 0 corresponds to perfect symmetry (i.e. the transition was experienced equally often in both directions), whereas an asymmetry index of 1 corresponds to maximal asymmetry (i.e. the transition was only ever experienced in one direction). Across subjects and transitions, there was large variability in the asymmetry index (*Figure 3B*).

We could exploit this variability to test for non-directionality in the neural signature, which is a feature of a map-like structure. We converted the number of times each transition was experienced into a distance measure for each individual subject according to the following equation:

$$d = 1 - \frac{c}{1 + c_{max}}$$

Here, $d$ denotes the length of the shortest path between two connected objects. It is computed based on the number of times this particular transition was experienced during training ($c$) relative to the number of times the most visited transition was experienced ($c_{max}$). The length of the path between objects that were two or three links away was then computed as the single-source shortest path between these objects (by adding the path-lengths for connected objects linking these two objects and choosing the shortest one). To compute the 'symmetric shortest path measure', the directional path-length measures (e.g. 5–2 and 2–5) were averaged. The directional and the symmetric shortest path measures were used as regressors to predict the neural signal extracted from the peak voxel in ROI 3 (*Figure 3C*). To assess the significance across subjects, we performed two-tailed paired t-tests on the regression coefficients.

To visualise the representation of the graph structure in the entorhinal cortex, we performed MDS on the neural activity extracted from the same peak voxel (ROI 3). MDS arranges objects

spatially such that the distances between them in space correspond to their similarities as defined by the distance matrix as well as possible. Here, we estimated the configuration of objects in two dimensions using the corresponding inbuilt Matlab function 'mdscale'. Specifically, MDS was performed on a matrix denoting the mean neural activity across subjects for each pair of transitions. For example, element 2–5 in the matrix corresponded to the average activity across subjects on trials where object 5 was preceded by object 2, and element 5–2 corresponded to the average activity across subjects on trials where object 2 was preceded by object 5. Because neural activity scales with distance, this matrix effectively corresponds to a distance or similarity matrix. Note that MDS can only be performed on symmetric matrices with positive entries. We therefore normalised the matrix by subtracting the minimum value of the matrix and adding 1, and then symmetrised it by averaging the top and the bottom triangles.

We tested the map-like representation for significance by comparing the Euclidian distances resulting from projecting our raw data into a two-dimensional space using MDS to a null distribution of graph structures generated by permuting the links. Specifically, the null distribution was generated by keeping the nodes in the location identified by the MDS, but then permuting the seven links making up the graph structure to random nodes. Only complete graphs were included in the null distribution, that is, graphs where each node was connected to each other node, either directly or indirectly. This results in 68,295 unique graphs. We then computed link distances for each graph and correlated the resulting link distance measure with the distances resulting from performing MDS on the average fMRI response. This provided us with a null distribution to which we could compare the correlation between the actual graph's link distance and the MDS-mapped data *Figure 3—figure supplement 1A*.

As a second test of the mapping, we computed the number of line crossings in this null distribution. A two-dimensional map is characterised by the fact that there are no line crossings between pairs of directly connected nodes. In the null distribution, this is only true for 13.17% of all graphs *Figure 3—figure supplement 1B*).

We repeated all analyses reported in *Figure 3* for parameter estimates extracted from an anatomically defined ROI comprising the entorhinal cortex and subiculum (*Figure 3—figure supplement 3*).

## Behavioural experiment

A separate group of 26 subjects (aged 19–31 years, mean age ± standard deviation 24.9 ± 3.7 years, 10 males) participated in a behavioural version of the experiment. Day 1 of the behavioural experiment was designed to be the same as day 1 of the fMRI experiment, with subjects performing 10 (n = 14) or 12 (n = 12) blocks of the object orientation cover task on object sequences generated according to a random walk along the graph structure. On day 2, subjects performed the same object orientation cover task on the reduced set of objects presented to subjects participating in the fMRI experiment in the scanner. Trial transitions were pseudo-randomised to ensure that each object was preceded by each other object the same number of times. This enabled us to test for changes in response times with distance between objects on the graph. Subjects performed 10 blocks of test trials with self-paced breaks in between blocks, with 127 objects presented in each block. Thereby, each stimulus–stimulus transition was probed three times per block, or 30 times across the experiment as a whole.

All analyses were performed on log-transformed response times in order to normalise response time measures. To account for object-specific effects that are independent from any distance-dependent scaling, we subtracted the mean response time for each object and subsequently computed average demeaned response times for each of the 42 stimulus–stimulus transitions per block (object 1 preceded by object 2; object 1 preceded by object 3, ..., object 7 preceded by object 6). We averaged these measures across blocks to obtain one representative measure per transition and subject.

To test for scaling of response times with distance on the graph, we regressed communicability, link distance, and Euclidian distance for each of the 42 transitions onto subject-specific response time measures. The significance of the regression across subjects was assessed using two-tailed paired t-tests on the resulting regression coefficients (*Figure 5C*). To visualise the relationship between communicability and response times, we sorted the data according by communicability,

created seven bins with an equal number of transitions in each bin, and plotted the mean log response times across subjects for each bin (*Figure 5D*).

## Acknowledgements

We thank Neil Burgess, Alon Baram, Tim Muller, Zeb Kurth-Nelson, Philipp Schwartenbeck, Alexandra Constantinescu, Archy de Berker, and Helen Barron for discussions and comments on the manuscript. This work was supported by the Wellcome trust (4-year PhD studentship 097267/Z/11/Z to MMG, Senior Investigator Award to RJD, 098362/Z/12/Z, and Senior Research Fellowship 104765/Z/14/Z to TEJB), the Joint Initiative on Computational Psychiatry and Ageing Research between the Max Planck Society and University College London (RJD), and the James S McDonnell Foundation (JSMF220020372, TEJB). The Wellcome Trust Centre for Neuroimaging is supported by core funding from the Wellcome Trust (Strategic Award Grant 091593/Z/10/Z).

## Additional information

### Competing interests

TEJB: Senior editor, *eLife*. The other authors declare that no competing interests exist.

### Funding

| Funder | Grant reference number | Author |
| --- | --- | --- |
| Wellcome Trust | 4-year PhD studentship, 097267/Z/11/Z | Mona M Garvert |
| Wellcome Trust | Strategic Award Grant to the Wellcome Trust Centre for Neuroimaging,091593/Z/10/Z | Mona M Garvert Raymond J Dolan Timothy EJ Behrens |
| Wellcome Trust | Senior Investigator Award, 098362/Z/12/Z | Raymond J Dolan |
| Max Planck Society | Joint Initiative on Computational Psychiatry and Ageing Research | Raymond J Dolan |
| University College London | Joint Initiative on Computational Psychiatry and Ageing Research | Raymond J Dolan |
| Wellcome Trust | Senior Research Fellowship, 104765/Z/14/Z | Timothy EJ Behrens |
| James S. McDonnell Foundation | JSMF220020372 | Timothy EJ Behrens |

The funders had no role in study design, data collection and interpretation, or the decision to submit the work for publication.

### Author contributions

MMG, RJD, TEJB, Conceptualization, Data curation, Formal analysis, Funding acquisition, Investigation, Methodology, Writing—original draft, Project administration, Writing—review and editing

### Author ORCIDs

Mona M Garvert, http://orcid.org/0000-0002-8678-5536
Timothy EJ Behrens, http://orcid.org/0000-0003-0048-1177

### Ethics

Human subjects: All subjects gave written informed consent and the study was approved by the University College London Hospitals Ethics Committee.

# Additional files

## Major datasets

The following dataset was generated:

| Author(s) | Year | Dataset title | Dataset URL | Database, license, and accessibility information |
|---|---|---|---|---|
| Mona M Garvert, Raymond J Dolan, Timothy EJ Behrens | 2017 | Data from: A map of abstract relationalknowledge in human hippocampal-entorhinal cortex | http://dx.doi.org/10.5061/dryad.nk08s | Available at Dryad Digital Repository under a CC0 Public Domain Dedication |

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
