## [Decision Letter]

Thank you for submitting your article "A map of abstract relational knowledge in human entorhinal cortex" for consideration by *eLife*. Your article has been reviewed by two peer reviewers, and the evaluation has been overseen by a Reviewing Editor and Sabine Kastner as the Senior Editor. The following individual involved in review of your submission has agreed to reveal his identity: Christian F Doeller (Reviewer #2).

The reviewers have discussed the reviews with one another and the Reviewing Editor has drafted this decision to help you prepare a revised submission.

You will see that the reviewers feel the work has great potential but that substantial additional evidence must be developed to support your main conclusion. Given the nature of their recommendation, we request that you provide a written response to the comments below with a plan of how you would proceed and a time table of the effort involved in addressing these concerns. We will share your response with the reviewers and let you know if they find your plan acceptable.

Both reviewers found the topic of the work to be novel and important. Understanding how the brain might represent knowledge is currently unknown. The current experiment tested whether the temporal relationships between stimuli may be coded in a predictable manner according to their distance. Leveraging repetition suppression analyses, the main results suggest that distance may be represented as the overlap between items.

The main concerns that were brought up by the reviewers were that no behavioral data were included. In order to convince that the activation patterns were related to implicit knowledge, demonstrating implicit memory associations between these items is necessary. One suggestion is to analyze RTs during learning to see if there is behavioral priming that is evident. Without a behavioral measure of implicit knowledge, the reviewers feel the claims of the paper are not justified. Reviewer 2 notes "I think the paper would be significantly strengthened if the authors could demonstrate with behavioral data that participants actually learned the underlying graph structure (see e.g. Seger (Psych. Bull. 1994) and Reber (J Exp Psych Gen 1989) for reviews on behavioural indices for implicit learning). For example, the authors could investigate response times of the cover task performed during scanning: If subjects had learned the graph structure, then one would expect faster responses to items, which were preceded by an item one link away during training (i.e. transitions following the rule; e.g. 1 → 2), than for items which could not follow each other based on the graph (i.e. transitions breaking the rule; e.g. 4 → 2)."

A second concern was the use of a peak voxel from another study. Although this is good practice to avoid double dipping the particular study examined spatial coding (Chadwick) so it is not clear this is the best study to choose. Can the authors demonstrate the effects without using this peak voxel, from their own data? Or are the effects dependent on this link? Related to this, it is surprising that no effects were seen in the hippocampus. This should be directly addressed. Especially based on the study by Morgan et al. (J Neurosci, 2011), which also capitalizes on fMRI adaptation as a function of distance, one might have predicted similar effects in the hippocampus. The authors should discuss this.

Another concern was regarding whether the results were picking up on a pure sequence effect since the authors define distance only as the number of links between two positions on the graph (e.g. Shendan et al. Neuron 2003) rather than a spatial effect per se. A reviewer notes "However, if spatial coding principles are involved in representing implicit knowledge, then distance could also be defined as the Euclidean distance between positions in the graph. Importantly, the authors have a way of testing this with their design. Objects 4 and 5 are not directly connected on the graph, but could be connected through a single link. Since the hippocampal formation is known to code for Euclidean distance (Morgan et al., 2011; Howard et al., 2014; Spiers & Maguire, 2007) one might expect that transitions between the objects 4 & 5 might elicit greater repetition suppression than transitions between other objects, which are separated by two links." Related to this, the MDS analysis is very important, since it might provide the only evidence for a map-like representation but it is difficult to judge if this is indeed the case. Is there a way to test how well the reconstructed map corresponds to the actual graph? Could the authors test a fit to equally complex, but implausible control configurations of the graph?

Importantly, much of the conclusions follow from the reasoning that items close together in the graph organization should demonstrate fMRI adaptation. However, it does not appear that fMRI adaptation in these regions has been demonstrated through, e.g., consecutive repetitions of the same stimulus. This control condition would greatly strengthen the claims that the authors could make about their results.

In addition, is there a reason why representational similarity analyses were not included? This is not a requirement but it would seem all of the claims are about the similarity of the underlying representations and repetition suppression just seems like a way to test this that is one step removed from the hypothesis – something for the authors to consider.

---

## [Author Response]

*[…] The main concerns that were brought up by the reviewers were that no behavioral data were included. In order to convince that the activation patterns were related to implicit knowledge, demonstrating implicit memory associations between these items is necessary. One suggestion is to analyze RTs during learning to see if there is behavioral priming that is evident. Without a behavioral measure of implicit knowledge, the reviewers feel the claims of the paper are not justified. Reviewer 2 notes "I think the paper would be significantly strengthened if the authors could demonstrate with behavioral data that participants actually learned the underlying graph structure (see e.g. Seger (Psych. Bull. 1994) and Reber (J Exp Psych Gen 1989) for reviews on behavioural indices for implicit learning). For example, the authors could investigate response times of the cover task performed during scanning: If subjects had learned the graph structure, then one would expect faster responses to items, which were preceded by an item one link away during training (i.e. transitions following the rule; e.g. 1 → 2), than for items which could not follow each other based on the graph (i.e. transitions breaking the rule; e.g. 4 → 2)."*

We agree with the critique and thank the reviewers for suggestions on how to address this shortcoming. While we did not have enough power to detect an implicit effect using the behavioural data collected in the scanner (subjects only performed a behavioural response on 1/10 of the trials, or 42 trials/block), we now included additional behavioural data to demonstrate that subjects do indeed acquire implicit knowledge about the graph structure.

In a separate group of 26 subjects we show that the distance between objects within the underlying graph structure is reflected in subjects’ response times as suggested by the reviewer. Subjects were trained on a speeded orientation judgement cover task in exactly the same manner as subjects participating in the neuroimaging study. They returned the next day to perform the same task on the seven objects presented to the neuroimaging cohort in the scanner, with random object transitions. We observed that, across subjects, response times in response to a given stimulus scaled with the distance on the graph between the currently presented object and the object preceding it in the sequence.

Notably, the distance metric that best explains response times in this context does not correspond to the link distance between pairs of objects (i.e. the shortest path), or the Euclidian distance between objects on the graph. Instead, response times are better explained by a distance measure we refer to as ‘communicability’, even in a situation where link distance and Euclidian distance are added as additional regressors in a multiple linear regression (communicability: t_25_ = 2.77, p = 0.01; link distance: t_25_ = -0.40, p = 0.69, Euclidian distance: t_25_ = -0.85, p = 0.40).

Communicability is a graph-theoretic measure capturing the distribution of future states in a graph. It is closely related to the successor representation, a predictive encoding scheme of states in a reinforcement learning world (Dayan, 1993). Recent theoretical accounts of hippocampal function suggest that such a successor representation may be encoded by hippocampal place cells (Stachenfeld et al., 2016, 2014). Both measures are predictive and reflect topological features of a graph (unlike Euclidian distances), which becomes particularly pertinent in situations where geometries are warped around obstacles in (physical or abstract) space.

In Figure 6, this novel distance measure is characterised by specific distortions in comparison to link distance and Euclidian distance measures. For example, transitions that form part of many paths around the graph structure are shortened (e.g. link 3-4) and links that are less frequently visited are substantially longer (e.g. link 6-4). We elaborate further on this novel measure in our fourth response below.

Author response image 1.Visualisation of communicability coordinates for the graph structure by performing multi- dimensional scaling on the communicability matrix.**DOI:**
http://dx.doi.org/10.7554/eLife.17086.016

We now added the following text to the Methods section:

“A separate group of 26 subjects participated in a behavioural version of the experiment. […] To visualise the relationship between communicability and response times, we sorted the data according by communicability, created 7 bins with an equal number of transitions in each bin, and plotted the mean log response times across subjects for each bin (Figure 5).”

We added the following text to the Results section of the manuscript:

“A map-like representation in the hippocampal-entorhinal cortex suggests that subjects acquired implicit knowledge about the graph structure, even in the absence of explicit awareness of any regularities in the object sequence. […] In line with our imaging results, response times did not scale with link or Euclidian distance between objects, but instead with communicability (communicability: t_25_ = 2.77, p = 0.01; link distance: t_25_ = -0.40, p = 0.69, Euclidian distance: t_25_ = -0.85, p = 0.40, Figure 5).”

*A second concern was the use of a peak voxel from another study. Although this is good practice to avoid double dipping the particular study examined spatial coding (Chadwick) so it is not clear this is the best study to choose. Can the authors demonstrate the effects without using this peak voxel, from their own data? Or are the effects dependent on this link?*

We appreciate this observation, although we consider it remarkable that the effect can be demonstrated in a peak voxel taken from a completely independent study investigating similar concepts in the spatial domain. More importantly, we would wish to retain the Chadwick voxel to avoid the pressing imaging concern of double dipping. Neither the link distance versus time nor the symmetry can be tested in any other ROI, because all other ROIs were selected for a symmetric link distance effect. Any such test would therefore be biased. Notably, this also includes the reconstruction of the map.

However, to alleviate any pressing concerns regarding the particular choice of ROI we have now added a figure supplement displaying the same analyses performed on data extracted from the anatomically defined ROI encompassing the entorhinal cortex and the subiculum displayed in Figure 2—figure supplement 1. The distance-dependent scaling of activity as well as all tests reported in Figure 3 are also significant if analyses are performed on data extracted from this second independent ROI (Figure 3—figure supplement 2).

We have now added Figure 3—figure supplement 2 and the following text to the Results section:

“Notably, the data were extracted from an independent ROI taken from an experiment investigating maps in allocentric physical space (Chadwick et al., 2015, ROI 3). Results are comparable if parameter estimates are extracted from an anatomically defined ROI comprising the entorhinal cortex and the subiculum (Figure 3—figure supplement 2).”

Text added to the Methods section:

“We repeated all analyses reported in Figure 3 for parameter estimates extracted from an anatomically-defined ROI comprising the entorhinal cortex and subiculum (Figure 3—figure supplement 2).”

*Related to this, it is surprising that no effects were seen in the hippocampus. This should be directly addressed. Especially based on the study by Morgan et al. (J Neurosci, 2011), which also capitalizes on fMRI adaptation as a function of distance, one might have predicted similar effects in the hippocampus. The authors should discuss this.*

We agree with the reviewers and we had no strong prior hypothesis about the precise spatial location, other than an expectation that we would find the map in the hippocampal-entorhinal system. Indeed, if we lower the statistical threshold we can see that activity extends into the hippocampus (Link distance, Figure 2, Figure 7 p < 0.05).

Author response image 2.**DOI:**
http://dx.doi.org/10.7554/eLife.17086.017

More importantly, however, using the new distance metric we introduce (the successor representation) we do find hippocampal activity, in particular if Euclidian distances are regressed out (Figure 4).

We elaborate on this in our fourth response below. We also toned down strong statements regarding the precise special localisation of the effect throughout the paper.

*Another concern was regarding whether the results were picking up on a pure sequence effect since the authors define distance only as the number of links between two positions on the graph (e.g. Shendan et al. Neuron 2003) rather than a spatial effect per se. A reviewer notes "However, if spatial coding principles are involved in representing implicit knowledge, then distance could also be defined as the Euclidean distance between positions in the graph. Importantly, the authors have a way of testing this with their design. Objects 4 and 5 are not directly connected on the graph, but could be connected through a single link. Since the hippocampal formation is known to code for Euclidean distance (Morgan et al., 2011; Howard et al., 2014; Spiers & Maguire, 2007) one might expect that transitions between the objects 4 & 5 might elicit greater repetition suppression than transitions between other objects, which are separated by two links."*

We would like to point out that the previous version of the manuscript already contained two analyses we think speak against a sequence effect. These include (1) the analysis that shows that distance is a better predictor than time (Figure 3) and (2) the analysis that shows that symmetrised distance is a substantially better predictor than the actual sequence distance that the subject experienced (Figure 3).

Nevertheless, we agree with the reviewers that it would be interesting to know whether the relationships between objects on the graph are actually mapped into a Euclidian space (“as the crow flies”). We also believed that the tests the reviewers proposed could identify such a mapping, however we could not find any significant difference for 4-5 and 5-4 relative to 2-5 and 1-4 transitions, which are equally far away in terms of the number of links, but further in terms of Euclidian distance (t_22_ = 1.61, p = 0.12).

In a direct comparison with Euclidian distances, we find that the fMRI adaptation pattern as well as response times are instead more consistent with distance measures reflecting the distribution of future states. One example for such a predictive representation of states is the successor representation facilitating the rapid computation of values in a reinforcement learning world (Baram et al., 2017; Dayan, 1993; Momennejad et al., 2016; Russek et al., 2016). Recent theoretical accounts of hippocampal function suggest that hippocampal place fields may reflect such a distribution of future states, rather than an animal’s current location in space (Stachenfeld et al., 2016, 2014). The successor representation provides a mathematical account of distances that easily generalizes to higher-dimensional, discrete and non-physical spaces and reflects geometric layouts such as topological distortions.

We believe that, together, these novel analyses and data provide evidence for a graph structure that is not Euclidian, but instead reflect a predictive representation of states, and added this new theoretical interpretation of our data to the manuscript.

We now added a sentence to the Abstract to reflect this new theoretical interpretation:

“Notably, the measure that best predicted a behavioural signature of implicit knowledge and BOLD adaptation was a weighted sum of future states akin to the successor representation that has been proposed to account for place and grid-cell firing patterns.”

We added the following section to the Introduction:

“We found no evidence for a mapping of discrete relationships into Euclidian space. […] It has recently been demonstrated that the successor representation can account for a number of properties of place cell and grid cell activity (Stachenfeld et al., 2016, 2014).”

We added Figure 4, Figure 4—figure supplement 1 and the following text to the Results section:

“In the reinforcement learning literature it has been suggested that a cognitive map of the relationship between states may be most useful if the representation of a state is predictive in nature and reflects the distribution of likely future states. […] Instead, these results are consistent with the view that the distance effect we observe in this system may be a consequence of the hippocampal formation encoding a predictive representation of states within a graph structure (Stachenfeld et al., 2016, 2014).”

We have added the following text to the Discussion:

“Notably, the resulting map is not Euclidian in nature. Instead, we find that the neural data as well as the behaviour are consistent with a measure corresponding to a weighted sum of future states in the graph structure. […] Such predictive representations of the relationships between discrete objects or states of the world may be combined with reward representations to enable flexible goal- directed behaviour (Baram et al., 2017; Dayan, 1993; Momennejad et al., 2016; Russek et al., 2016).”

*Related to this, the MDS analysis is very important, since it might provide the only evidence for a map-like representation but it is difficult to judge if this is indeed the case. Is there a way to test how well the reconstructed map corresponds to the actual graph? Could the authors test a fit to equally complex, but implausible control configurations of the graph?*

We would like to thank the reviewers for this important suggestion. We now performed permutation tests on the map by comparing the correlation between Euclidian distances resulting from placing nodes in a 2-dimensional space using multi-dimensional scaling and link distances of the actual graph structure to a null-distribution.

We added the following text and to the Results section:

“Permutation tests confirm that the MDS-mapped distances are significantly more correlated with link distances of the original graph structure than with link distances of a null distribution consisting of all other complete graphs with 7 links (r = 0.65, p = 0.003, Figure 3—figure supplement 1). Furthermore, no links cross in the graph resulting from the MDS mapping. This is only true for 13.17% of all possible graphs with nodes in the same location, but 7 randomly distributed links (Figure 3—figure supplement 1).”

We added a new Figure 3—figure supplement 1 to display the results of the permutation tests.

We added the following text to the Methods section:

“We tested the map-like representation for significance by comparing the Euclidian distances resulting from projecting our raw data into a 2-dimensional space using multi-dimensional scaling to a null distribution of graph structures generated by permuting the links. […] A 2-dimensional map is characterised by the fact that there are no line crossings between pairs of directly connected nodes. In the null-distribution, this is only true for 13.17% of all graphs.”

*Importantly, much of the conclusions follow from the reasoning that items close together in the graph organization should demonstrate fMRI adaptation. However, it does not appear that fMRI adaptation in these regions has been demonstrated through, e.g., consecutive repetitions of the same stimulus. This control condition would greatly strengthen the claims that the authors could make about their results.*

To increase power for the effects of interest we did not collect any data on stimulus repetitions in our experiment, so that we cannot make any claims about fMRI suppression effects in response to a repeated stimulus in the entorhinal cortex. However, electrophysiological recordings of entorhinal cells demonstrate that entorhinal neurons suppress in response to repeated stimulus presentation (Xiang and Brown, 1998). Furthermore, a hexagonal symmetry characteristic for entorhinal grid cell firing has also been demonstrated in a study investigating grid cells in human navigation using Fmri adaptation (Doeller et al., 2010). This suggests that entorhinal (grid) cells indeed adapt in response to repeated activation, which can be measured using fMRI adaptation.

Independent from this point, however, we would like to point out that the interpretation of our results does not rely on suppression effects in response to repeated presentation of the same stimulus. In fact, we believe it is an intriguing question as to whether an object that is embedded in a graph will show repetition suppression to itself in the entorhinal cortex. Whilst we agree it is interesting, we are not sure why this effect would be any more convincing than the three vs. two or two vs. one conditions that we have already shown. Unfortunately, we do not have the data to investigate this question. However, we now discuss this point and suggest future relevant experiments in the Discussion.

*In addition, is there a reason why representational similarity analyses were not included? This is not a requirement but it would seem all of the claims are about the similarity of the underlying representations and repetition suppression just seems like a way to test this that is one step removed from the hypothesis – something for the authors to consider.*

We agree with the reviewers that representational similarity analyses would be an equally valid, and potentially more direct, approach for providing insight into a similarity in neural representations. In this case, our experimental design was optimized for fMRI suppression analyses in terms of experimental variables such as the duration of inter-trial intervals. One reason to do this was a concern regarding the spatial extent of the similarity effect we hypothesized to find within the hippocampal-entorhinal cortex. Unlike suppression analyses, RSA relies on distributed patterns across voxels, and thus requires a pattern that varies across a reasonably sized ROI. Suppression effects, on the other hand, can in theory be detected within single voxels. We believed this may provide more sensitivity in the absence of precise estimates of the spatial extent of a neural effect.

However, we agree with the reviewers that RSA has a number of advantages, including a potentially more direct measure of similarity patterns across voxels, and we are working on similar approaches in the future.